# Experimental Evaluation of Shrinkage, Creep and Prestress Losses in Lightweight Aggregate Concrete with Sintered Fly Ash

**DOI:** 10.3390/ma14143895

**Published:** 2021-07-13

**Authors:** Rafał Stanisław Szydłowski, Barbara Łabuzek

**Affiliations:** Department of Civil Engineering, Cracow University of Technology, Warszawska 24 Street, 31409 Cracow, Poland; b.labuzek@gmail.com

**Keywords:** creep, lightweight aggregate concrete, prestress loss, shrinkage, sintered fly ash

## Abstract

The paper presents the experimental results of shrinkage, creep, and prestress loss in concrete with lightweight aggregate obtained by sintering of fly ash. Two concrete mixtures with different proportions of components were tested. Concrete with a density of 1810 and 1820 kg/m^3^, and a 28-day strength of 56.9 and 58.4 MPa was obtained. Shrinkage and creep were tested on 150 × 250 × 1000 mm^3^ beams. Creep was tested under prestressing load for 539 days and concrete shrinkage for 900 days. The measurement results were compared with the calculations carried out according to the Eurocode 2 as well as with the results of other research. A very low creep coefficient and lower shrinkage in relation to the calculation results and the results of other research were found. It was also revealed that there is a clear correlation between shrinkage and creep, and the amount of water in the concrete. The value of the creep coefficient during the load holding period was 0.610 and 0.537, which is 56.0 and 49.3% of the value determined from the standard. The prestressing losses in the analyzed period amounted to an average of 13.0%. Based on the obtained test results, it was found that the tested lightweight aggregate concrete is well suited for prestressed concrete structures. Shrinkage was not greater than that calculated for normal weight concrete of a similar strength class, which will not result in increased loss of prestress. Low creep guarantees low deflection increments over time.

## 1. Introduction

The first application of lightweight concrete as a construction material is known from ancient times when, due to its benefits derived from a lower density and simultaneously high strength parameters, lightweight concrete was used to erect buildings for which construction with the use of heavier materials was hampered. It was mainly used when large spaces had to be covered or when there was a need to reduce the load imposed on land.

The use of lightweight concrete can be traced to as early as 3000 BC, when Mohenjo-Daro and Harappa were built during the Indus Valley civilization [1]. However, in Europe, its first use occurred 2000 years ago when the Romans built the Pantheon, the aqua ducts, and the Colosseum in Rome [2]. Some of these magnificent ancient structures still exist, such as the St. Sofia Cathedral or Hagia Sofia, in Istanbul, Turkey, built by two engineers, Isidore of Miletus and Anthemius of Tralles, and commissioned by the Emperor Justinian in the 4th century A.D. However, the use of lightweight concrete was limited after the fall of Roman Empire, until the 20th century when a new type of manufactured material named expanded shale, which is a lightweight aggregate, became available for commercial use.

The new beginning of lightweight concrete in modern times dates to 1917, when S. J. Hyde developed a rotary kiln for drying slate and clay, thus obtaining a material lighter than traditional concrete. At around the same time, F. J. Straub pioneered in the use of coal ash in the production of concrete blocks employed in the construction of buildings [3]. The material was first used on a massive scale in 1918 during World War I for the construction of ships and barges. The US fleet concluded that concrete with a maximum density of 1760 kg/m^3^ and a compressive strength of no less than 28 MPa would be an effective material.

Currently, due to a number of advantages, lightweight concrete is used both in prefabrication and in cast-in-situ concrete construction. The lower density paired with high strength allows for the reduction of cross-sections, increasing the span or service loads. Lightweight concrete has been used in monolithic construction in the world, primarily in high-rise buildings (reducing their weight), in long-span roofs (reducing the dead load, which in the case of self-weight is the leading load [4]), in multistorey car parks, in bridges of various, often complex architectural form, and liquid containers [5]. In prefabrication, lightweight concrete is used for the production of columns, beams, floor slabs, and walls, as well as girders and bridge decks.

Examples of the application of lightweight concrete in high-rise buildings are Water Tower Place in Chicago, Yokohama Landmark Tower in Japan, Commerzbank Tower in Germany, and the Shard in London, which was once the tallest building in the European Union.

A different field of concrete construction is prestressed concrete structures. Here, too, lightweight concrete has been used for several dozen years, mainly in bridges, allowing for achievement of larger spans and smaller cross-sections. As has been shown by many studies [6,7], lightweight concrete behaves much better under variable and dynamic loads than normal weight concrete. The lighter weight of the structure ensures higher natural frequencies, lower vibration amplitudes, and higher damping. One of the first prestressed bridges was built in 1978 in California over New Melones Lake, where lightweight concrete was used to reduce the weight of the structure. The bridge is designed with a rectangular cross-section and has a span of 195 m. Another lightweight concrete bridge with the world’s largest beam span, a length of 301 m [8], was built in 1998. It connects the islands of Stolmen and Selbjørn in Norway. Such a large span was made possible through the use of a variable rectangular cross-section made from lightweight concrete. In 2005, California saw the construction of the Benicia–Martinez Bridge [9]. The choice of lightweight concrete contributed significantly to a reduction in the bridge construction costs. The bridge has a total length of 2.4 km and 22 spans ranging from 127 to 201 m, of which 16 spans are located above the water.

Long-span structures made of lightweight concrete also include the Doncaster Racecourses Grandstand and Wellington Regional Stadium in New Zealand, the Oberstdorf ski flying hill, and the Bergsøysundet and Nordhordland pontoon bridges. It is also worth mentioning the highest structure in the world to have been transported, i.e., the Troll gas platform, which was made entirely of lightweight concrete [8].

Unfortunately, lightweight concrete has not been used in the construction of posttensioned concrete long-span slabs so far. The authors of this paper took a few years to design several post-tensioned concrete slabs with unprecedented spans and span-to-depth ratio [10,11,12]. Normal weight concrete was used for the slabs. However, numerous computational analyses [4,13] have shown that the use of lightweight concrete may be more favorable. The underestimated modulus of elasticity in such concrete is overcompensated by the lower weight of the slab. Appropriate prestressing can allow the concrete to obtain smaller final deflections than in the case of heavier normal weight concrete slabs. This can enable the construction of larger and more slender slabs than before.

The rheology of concrete is of great importance in thin post-tensioned concrete floor slabs with a span-to-depth ratio of more than 40. On the one hand, increased shrinkage and creep of concrete will lead to greater prestress losses, while on the other, they will cause excessive increases in long-time deflections. Because of the limited scope of the research carried out in this area so far and the lack of sufficiently precise standard procedures that describe these phenomena in lightweight concrete, the authors carried out long-term tests of concrete shrinkage and creep on the new Polish sintered fly ash aggregate Certyd. The aim of the research is an initial identification of the rheological characteristics of concrete before using it for long-span post-tensioned floor slabs. For this purpose, two concrete mixtures of different compositions were made. Nine beams with dimensions of 150 × 250 × 1500 mm^3^ were made of the mixtures. Some of the beams were post-tensioned 16 days after concreting. The remaining beams were unloaded as witness specimens. All specimens were placed in an air-conditioned chamber. The prestressing was released after 539 days. The strains were measured on all specimens for 900 days. Witness specimens were used to determine concrete shrinkage. The difference between the average strain of prestressed beams and witness beams allowed the researchers to determine the development of creep. The release of the prestressing made it possible to determine the immediate and delayed strain return of the concrete and the value of irreversible strain.

## 2. Background

### 2.1. Properties of Lightweight Concrete

Lightweight aggregate concrete is a composite with a matrix and aggregate that, unlike normal weight concrete, have a similar modulus of elasticity. This results in a more even distribution of stresses in the concrete structure, and thus a lower likelihood of crack propagation and increased concrete durability. Another advantage of lightweight aggregate concrete structures is the tight construction of the matrix-aggregate contact zone and the regular shape of grains in artificial aggregates, which is reflected in the high strength–density ratios. The structure of the contact zone between the aggregate and cement matrix, different in relation to concrete with rock aggregates, causes a different behavior of lightweight concrete under load and exhibits a different collapse mechanism. Lightweight concrete is not associated with the occurrence of three stages of crack development, as is the case in normal weight concrete (I—formation of stable features, II—stable crack propagation, III—unstable crack propagation). For typical normal weight concretes with rock aggregate, stage I goes into II at compressive stresses amounting to 30–40% of concrete strength, while stage II goes into stage III at stresses of about 70–90% of strength. In lightweight aggregate concrete, the first load-induced cracking appears only at stresses of 85–90% of strength [14]. It was ascertained in [15] that, for concretes with sintered ash aggregate, the straight-lined course of the dependence σ–ε extends up to 90% of the strength. The high elastic energy stored as a result of such behavior causes rapid crack propagation, which irreversibly leads to sudden destruction of the material.

Figure 1 shows the scheme of destruction in normal and lightweight concrete. Figure 1a shows the scheme of destruction in tension, whereas Figure 1b shows destruction in splitting. In normal weight concrete, the destruction usually occurs in the contact zone (aggregate/cement matrix), which is the weakest link in the concrete structure and, at the same time, the most loaded. In this zone there is a stress concentration caused by a considerable difference in the modulus of elasticity of the matrix and the aggregate. The different properties of the aggregates and matrix in normal concrete cause the destruction by separating the matrix from the aggregate. In lightweight aggregate concrete, the modulus of elasticity of the matrix and the aggregate are more similar than in normal weight concrete. This property leads to more uniform distribution of stresses, simultaneously reducing the stress concentration whereby the destruction occurs at the weakest element of the structure, which is the aggregate.

The study [16] showed that, in the case of lightweight concretes, the strength is influenced by the same properties as in the case of normal weight concretes, i.e., W/C ratio, cement content, and age of concrete. Therefore, to obtain the same class with the same volumetric composition of lightweight and normal weight concrete, a matrix with higher strength should be used in lightweight concrete.

An important parameter of structural concrete is the modulus of elasticity. Due to the nature of concrete, which is a composite of aggregate and matrix, the modulus of elasticity depends on the modulus of both components, taking into account their volumetric contributions and mutual adhesion. Similarly to normal weight concrete, in lightweight concrete the modulus of elasticity largely depends on the aggregate; however, here, the aggregate is the weaker link, which causes a considerable reduction in the modulus of the entire composite. In paper [17], it was shown that, in the general case of lightweight concrete, the modulus of elasticity may be 15–60% lower relative to normal weight concrete of the same strength classes, depending on the density of the concrete and the aggregate used.

The different structure of concrete with lightweight aggregate, in addition to lower values of strength properties and increased shrinkage, causes a different creep range compared to normal weight concrete. Many publications have indicated that structurally lightweight aggregate concretes may show greater creep than those with rock aggregates of comparable strength classes. Report BE 96-3942/R2 [18] stated that the creep strain may be 20–60% higher compared to concretes with normal weight aggregates. The dynamics of creep in lightweight concrete over time are also greater. However, this statement was based on older results of tests conducted on concretes of relatively low strength. Meanwhile, as is well known, the higher the strength of concrete, the lower the creep. Only structural lightweight concretes of lower strength (up to 20–30 MPa) can have a slightly higher creep value compared to normal weight concretes. Paper [19] shows that a strength increase in concrete with lightweight fine and thick aggregate from 20.7 to 34.5 MPa causes a 20–40% decrease in creep. Lightweight concretes of higher strength, especially those of high strength, show similar and sometimes even lower creep compared to concretes with normal weight aggregates [20,21,22,23]. This is possible because concrete creep is determined by the creep of cement slurry. In lightweight concrete, the matrix is usually characterized by higher strength compared to the matrix of normal weight concrete of the same class. As a result, although the less rigid lightweight aggregate cannot inhibit the creep strain of the cement matrix as effectively as ordinary aggregate, the matrix creep itself is smaller in lightweight concrete.

Thermal treatment has a beneficial effect on the reduction of lightweight concrete creep. As a result of low-pressure machining, lightweight concrete creep can be reduced by 25–45% compared to concrete subject to wet treatment. Use of autoclaving turns out to be even more effective—it is possible to reduce creep even by 60 to 80% [19].

### 2.2. Creep and Shrinkage Research

Shrinkage of lightweight aggregate concrete has been the subject of many studies [23,24,25,26,27] which showed a much higher (even by 50%) shrinkage of such concretes compared to normal weight concrete of a similar strength class. Unlike shrinkage, lightweight concrete creep has been the subject of very few studies so far. The issue of creep testing is exacerbated by the high cost of maintaining constant stresses in the long-term, in constant thermal and humidity conditions. The results obtained in a short period of time are difficult to interpret and do not allow for any conclusions on the extent of the final creep. To date, there have been no uniform regulations regulating the creep test methodology; therefore, the test results obtained on samples of various sizes and at different stress levels do not show a clear picture of the rheological quality of lightweight concrete in relation to normal weight concrete. The creep tests for lightweight aggregate concrete available in the literature along with the most important parameters are summarized in Table 1.

Results of one of the first lightweight concrete creep tests were published by Best and Polivka in 1959 [28]. The authors studied concrete with aggregates from baked shale, with 28-day strength of 20.7 and 34.5 MPa. After 520 days of loading, they found that lightweight concrete creep is similar or smaller compared to concrete made of gravel aggregate of similar strength.

The study [29] investigated the shrinkage and creep of lightweight concrete with various lightweight aggregates (expanded blast furnace slag, expanded shale produced in a rotary kiln, expanded shale produced on sintering grate, expanded clay produced on sintering grate) at different levels of sand in concrete. Lightweight aggregate was replaced with sand in the amount of 0, 33.3, 66.7, and 100%. The tests were carried out on cylindrical specimens φ150 × 300 mm^2^, loaded for 730 days. It was shown that both shrinkage and creep decreased with increasing sand content. For lightweight concretes containing 0 to 100% sand, a creep coefficient of 1.26 to 1.00 was obtained. When the lightweight aggregate was completely replaced with sand, the creep coefficient was up to 30% lower compared to concretes with other proportions of sand and lightweight aggregate.

Research published in [30] studied the shrinkage and creep of concrete with sintered slate aggregate, with an average final strength of 68.5 and 75.4 MPa and a density of 1875 and 1905 kg/m^3^, respectively. A total of 26 samples from two concretes were tested. The load was applied after 16 and 24 h from concreting to the level of 40 and 60% of the current compressive strength. Six hundred and twenty-day values of the shrinkage and creep coefficient were determined. Based on the conducted tests, it was found that, for the higher strength mixture, the creep after 620 days is lower and, in this case, the age at the moment of loading is of little importance. The shrinkage study showed that for both mixtures, 90% of the shrinkage measured after 620 days had occurred after 260 days. 

The paper [31] presents tests of creep and shrinkage of elements on a natural scale (prestressed concrete beams) and small specimens made from lightweight concrete with a compressive strength exceeding 55.2 MPa. Pre-soaked extended shale was used as lightweight aggregate. Except for natural-scale beams, creep was tested on cylindrical specimens of φ100 × 380 mm^2^, φ150 × 300 mm^2^, and beams 38 × 38 × 125 mm^3^. Samples were subjected to a constant load after 24 h and 28 days of maturing of the concrete; furthermore, measurements of strains caused by creep were taken for 120 days. It was observed that creep increased with decreasing specimen size. The authors also observed less creep for lightweight concrete compared to normal weight concrete. This effect was justified by the presence of absorbed water in lightweight concrete samples due to the use of pre-wetted aggregate. It should be emphasized, however, that in the presented research, shrinkage and creep strains were not separated. All conclusions for creep are based on the analysis performed by DIC (Digital Image Correlation Technique).

The results of more recent creep studies on self-compacting lightweight aggregate concrete were published in 2018 in [32]. Concrete made with aggregate produced from sintered clay shale was tested. However, the resulting concrete was heavier than it is typically for this type of aggregate. Dry density was 1999 kg/m^3^. Seven cylindrical specimens with a diameter of 100 mm and a height of 1245 mm were tested. At the same time, the shrinkage was measured on four cylindrical test specimens of φ100 × 200 mm^2^. The first series (four cylinders) was loaded after one day with a stress of 14.8 MPa (which comprised 40 and 50% compressive strength after one day). The second series was loaded after 28 days with stresses of 19.3 MPa, which accounted for 40% of the compressive strength. The observations were carried out for a period of one year. Simultaneous tests were carried out on lightweight and normal weight concrete. It was noticed that the creep coefficient for lightweight concrete is slightly lower than for conventional concrete for a load time of one day; however, the intensity of its development in the first days of loading is higher for lightweight concrete. For a load applied after 28 days for lightweight concrete, a greater creep coefficient was obtained than for normal weight concrete.

The results of the unique creep studies were published in 2020 in the paper [33]. The authors of the work, for 30 years, tested 16 reinforced concrete and prestressed lightweight concrete beams under long-term load. Concrete with a density of 1800 kg/m^3^ was used. Two kinds of lightweight aggregate were used: expanded clay and sintered pulverized fuel ash. The beams had a span of 3.00 m, varied cross-section (T, inverted T, rectangular) with a height of 240 mm, varied ordinary and prestress reinforcement ratio, and varied compressive concrete strength (25.1–42.3 MPa). The presented results are difficult to compare with the creep results from other studies. The authors found that most of the creep effects accumulate in the first year, strong creep effects last for the first 5 years, and a noticeable incremental creep displacement exists through 20 and 30 years of testing.

In 2020, other but much poorer and poorly documented studies were also published [34]. The authors examined concrete prisms with dimensions of 150 × 150 × 600 mm^3^ and 50 × 150 × 450 mm^3^ made of lightweight concrete with expanded clay, expanded perlite, and agloporite. Concrete compressive (cubic) strength was from 10 to 60 MPa. The authors reported concrete deformation after two years, however, they did not draw any significant conclusions. The results were presented in a way that made it difficult to analyze and compare with others.

### 2.3. Research Gap

As mentioned earlier, the creep of concrete with lightweight aggregate has so far been the subject of few studies. Creep is an essential property of concrete for prestressed structures. The few studies conducted mainly included artificial aggregates obtained from natural resources (shales, clays, perlite, or agloporite). There is little research on concrete with aggregates produced from waste. Due to the high demand for natural aggregate at the beginning of the 21st century, the Polish Aggregates Producers Association predicts that in the next 10 years there will be no more sands and gravels, while the next 50 years will bring a shortage of crushed stone aggregates. Meanwhile, the presence in Poland of large amounts of waste in the form of ash from the production of electricity derived from the combustion of hard coal speaks for the production of artificial aggregate. The novelty of the presented research compared to the previous studies is the use of new Polish artificial aggregate with much better mechanical properties than the aggregates used previously, whence it could be assumed that the tested concrete is characterized by improved rheological properties. From here, it can be concluded that the use of artificial aggregates will have a positive impact on the rational management of natural resources.

## 3. Materials and Methods

### 3.1. Lightweight Aggregate

The few creep tests carried out so far, reported in Section 2, covered concretes with different types of artificial aggregates. In most cases, they were made of natural resources. They were expanded shales produced in a rotary kiln or sintering grate [28,29,30,31,32], expanded perlite, or agloporite [34]. In a few cases, they were waste materials such as expanded blast furnace slag [29] or sintered pulverized fuel ash [33].

This research was begun when the production of a new Certyd artificial aggregate was launched in Poland in 2015. It is produced with sintered fly ash. Certyd aggregate is produced from ash from heat and power plants deposited on heaps, which derive from the burning of hard coal. These are fly ashes from electrostatic precipitators and ash–slag mixtures from wet carrying of furnace waste. The aggregate is produced by sintering ash at a temperature of 1200 °C. The process occurs without the use of external fuel, using the heat from the combustion process of carbon residues in ash. Only a small amount of energy is needed at the beginning to the launch process. Lightweight aggregate in the form of regular spherical granules or crushed grains of various fractions is obtained from the remains after the combustion of hard coal (Figure 2). The basic parameters of the Certyd aggregate are given in Table 2 [35].

### 3.2. Concrete Mixtures

Two concrete mixtures were made with a dry density of 1810 and 1820 kg/m^3^. Cement CEM I 42.5 N was used. The amounts of individual components for the C-1 and C-2 mixtures are given in Table 3. The difference in the two mixtures was mainly the water content and the W/C ratio. The C-1 mixture contained 164 L of water per cubic meter and the C-2 mixture contained 209 L.

### 3.3. Test of Concrete Strength Properties 

As part of this work, the strength properties of two types of concrete, C-1 and C-2, were tested. The average compressive strength, modulus of elasticity, modulus of rapture, axial tensile strength, and splitting strength were tested after 7, 14, and 28 days. Each of the strength properties was tested on three specimens. Finally, we prepared nine cylindrical specimens φ150 × 300 mm^2^ for compressive strength testing, nine cylindrical specimens φ150 × 300 mm^2^ for modulus of elasticity testing, nine cylindrical specimens φ150 × 300 mm^2^ for axial tensile strength testing, nine beams 150 × 150 × 600 mm^3^ for modulus of rapture testing, and nine cubes 150 mm for splitting strength testing. The specimens were taken out from the mold and placed in water, then pulled from it just before testing.

### 3.4. Creep and Shrinkage Test

Nine beams with dimensions of 150 × 250 × 1000 mm^3^ (Figure 3) were made for the testing of shrinkage and creep (five from the C-1 mixture and four from the C-2 mixture). Some of them were loaded and some of them remained unloaded. The load was applied to prestressing tendons. Two 15.2 mm steel strands were used in each loaded beam. A special non-slip system of threaded anchorages was used. Ring dynamometers were installed under the anchorages to continuously monitor the force values in the tendons. Two beams from each mixture were prestressed after 16 days of concrete maturation (Figure 3). The values of stresses in individual beams just after anchoring the strands are given in Table 4. Initial stresses in the beams ranged from 9.0 to 11.0 MPa. The remaining beams (two from the C-1 mixture and three from the C-2 mixture) were unloaded and used to monitor the strain that resulted solely from shrinkage. Creep strains were determined by subtracting the strain of the unloaded beams from the strain of the loaded beams.

Two 200 mm long measuring bases were installed on both surfaces of each beam for measuring strain with a mechanical extensometer DEMEC. All beams (already on the second day after concreting) were placed on a steel frame in an air-conditioned chamber (Figure 4).

## 4. Results and Discussion

### 4.1. Concrete Strength and Modulus of Elasticity

The prepared specimens (cylinders, cubes, beams) were used to examine the strength characteristics of concrete after 7, 14, and 28 days of concrete maturation. Each feature was determined on three specimens. The average compressive strength and modulus of elasticity of concrete were determined on φ150 × 300 mm^2^ cylinders. Despite the different water content and W/C ratio (0.41 and 0.51), similar values of the average compressive strength were obtained after 28 days (Figure 5a), i.e., 56.9 MPa for C-1 and 58.4 MPa for C-2. Even lower compressive strength was achieved for lower W/C. This is explained by the insufficient amount of water needed for full hydration of the cement, which has been absorbed by the aggregate with a water absorption near 20%. In the case of both mixtures, the obtained compressive strength (with a slight deficiency in the case of the C-1 mixture) satisfied the preconditions of the LC50/55 class according to [36] (Table 5).

In the case of the modulus of elasticity for both mixtures, similar but low values were obtained (Figure 5b), i.e., 22.1 and 22.4 GPa. It is, respectively, 12.6 and 10.3% of the value required for class LC50/55 concrete, calculated for a density of 1810 kg/m^3^. However, when comparing with normal weight concrete of the corresponding strength class (C50/60), the obtained values are 40.3 and 39.5% lower, respectively (*E_cm_* = 37 GPa).

The axial tensile strength obtained in the tests (Figure 5d) is 3.86 and 3.48 MPa, and it is, respectively, 1.05 and 0.95 of the value required for the LC50/55 class (Table 5).

### 4.2. Air Condition

The beams used for shrinkage and creep tests were placed in an air-conditioned chamber and kept there for 900 days. A constant temperature of 20 °C and a humidity of 50% were programmed. Excluding the first day after demolding, the recorded temperature values in the chamber ranged between 18 and 23.7 °C, and the humidity varied from 45 to 54% (Figure 6).

### 4.3. Strains of Loaded and Unloaded Specimens

Figure 7 shows the course of the recorded strains in all nine beams. The strain value for each beam is the average of four measurement bases (Figure 3). Figure 7a shows the strains of the loaded beams and Figure 7b of the unloaded beams. After 555 days from concreting (539 days from loading), the load was removed.

Figure 8 shows the beam strains as average for both C-1 and C-2 concretes, loaded and unloaded. After 844 days, the shrinkage strains for unloaded beams were 385 and 514 με for C-1 and C-2 concretes, respectively. The 33% difference in the recorded shrinkage can be explained by the different water content in the mixture (164 and 209 l/m^3^). Immediate strain under loading was: for concrete C-1: 473 − 126 = 347, for concrete C-2: 525 − 119 = 406 με, while the strain recovery upon removing the load was 865 − 610 = 265 and 1122 − 764 = 358 με. Thus, the immediate strain recovery was 76% for C-1 concrete and 88% for C-2 concrete. Therefore, less water content in the concrete means less strain recovery during unloading.

Figure 8 also contains a shrinkage diagram determined according to the standard [36] for LC50/55 and C50/60 class concrete. The recorded shrinkage values are lower than the standard values determined for both concrete classes, especially for C-1 concrete with a smaller water content.

Figure 9 summarizes the measured shrinkage with other studies, on various lightweight aggregates, which are reported in Section 2.2. It can be seen that these are the longest of the presented studies on the shrinkage of lightweight aggregate concrete. Generally, in the period in which the comparison is possible, the analyzed concrete showed lower shrinkage than the previously tested concretes with artificial aggregate. Only C-1 concrete showed greater shrinkage in the first 500 days, compared to the concrete tested by Best and Polivka [28], made with baked shale. In the case of other studies presented, the shrinkage of the analyzed concrete is lower.

### 4.4. Strains Under the Load

Due to the fact that the loaded beams contained strains caused by both shrinkage and load, and the shrinkage of concrete is independent of the load, the difference in the average strains of the loaded and unloaded beams allowed for isolation of only the strains caused by the load. The development of these strains over time for both tested concretes is shown in Figure 10.

In general, concrete deformation over time under the load is divided into: instantaneous and time-dependent, as well as recoverable and recoverable. There are four kinds of deformation: elastic (instantaneous and recoverable deformation), plastic (instantaneous irrecoverable deformation), delayed elastic (time-dependent recoverable deformation), and viscous (time-dependent irrecoverable deformation). Elastic deformation is usually associated with the energy stored in the crystal or molecules that is fully recoverable. Plastic deformation occurs when a slip in a plane of maximum shear stresses changes positions of crystals, molecules, or atoms. Slip in the plane of maximum stresses presents no change in volume and it is not time-dependent. Delayed elasticity is usually a consequence of a lack of order in the microstructure, upon loading the microstructure slowly reaccommodates. The energy is not dissipated but stored, and it is therefore fully recoverable. Finally, viscous deformation describes the behavior of fluids and appears only under sustained load. The strain rate is proportional to the applied stresses, and there is no recovery upon load removal. The last two forms of deformation are considered to cause creep in concrete.

Based on the presented values of strains, four strain components discussed above were separated (Figure 11). The instantaneous strain was divided into an elastic (recoverable) and a plastic (irrecoverable) part. Delayed strain was divided into a delayed elastic (time-dependent recoverable) and a viscous (time-dependent irrecoverable) part. The obtained values along with their calculations are given in Figure 11.

It can be seen that the higher the water content in the concrete (C-2), the more elastic and the less plastic the behavior in the concrete. Higher water content also causes a more viscous behavior in the concrete. It is fully justified because viscous deformation describes the behavior of fluids.

The sum of plastic and viscous strains indicates residual strain. Residual strain values are 173 and 185 με adequate (Figure 10). It can be seen that the water content has little effect on the residual strain. However, a high level of residual strains is noticeable compared to instantaneous strain. It is 53.7% for concrete C-1 and 50.7% for C-2. This is due to the high values of viscous strains which proves the highly viscous behavior of the tested concrete.

### 4.5. Creep Coefficient

Figure 12 shows the development of the creep coefficient, defined as the ratio of delayed strain to immediate strain. The final creep coefficient (after 539 days of imposing the load) was 0.610 and 0.537. The same diagram also shows the creep coefficients determined according to the standard [36] for concrete LC50/55 and C50/60 (solid green and dashed line). The obtained values were 1.09 for lightweight concrete and 1.59 for normal weight concrete. It is easy to notice that the measured values of the creep coefficient are much lower than those determined from the standard. The values from the experimental studies are 56.0 and 49.3% of the calculated value. Referring the measured values to those for normal concrete, they account for 38.4 and 33.8%, respectively. The obtained values indicate a very low creep in the tested concrete, much lower than that provided for by the standard procedure.

The same figure compares the measured creep coefficient with the results of other presented tests of concrete with lightweight aggregates, in which the results were presented sufficiently for comparison. The measured creep coefficient for concrete with sintered fly ash is significantly lower compared to the results of all other presented tests.

### 4.6. Prestress Losses

Low relaxation steel tendons with a diameter of 15.5 mm were used to introduce the stresses into the loaded specimens. The strength of the steel was *f_pk_* = 1860 MPa and the strand cross-sectional area was 150 mm^2^. Figure 13 shows the course of the force in the prestressing tendons over time. The value of the initial force (after anchoring) was from 168.9 to 206.5 kN and the average value was 186.1 kN. It corresponds to the average stresses in the steel equal to 1241 MPa. It is 0.67 *f_pk_.* The wide dispersion in force results from the short length of the elements (1000 mm) and thus the major influence of anchoring on the change of force. Aalthough a special non-slip system of threaded anchorages was used, the short elements are sensitive to anchoring inaccuracies. The mean value of the force after 539 days of prestressing was 161.9 kN and the mean loss was 13.0%. The maximum force decrease (for the C-2/1 beam) was 16.2%. The difference in losses can be noticed between the beams made from C-1 and C-2 concrete. For beams made of the first mixture, losses of 11.5 and 10.9% were recorded, while for beams made of the second mixture—16.2 and 13.7%. This difference can be explained by higher shrinkage in the case of C-1 concrete caused by more water (Figure 8) and a higher creep strain (Figure 10).

It is commonly thought that the rheological prestress loss is no more than 10%, and such is considered acceptable. However, a value of 13% (or even 16%) does not discriminate against using concrete for prestressing and it can also be considered acceptable. It should be emphasized that these results were obtained on small-scale elements. In the case of elements on a larger scale, the prestress decrease may be smaller (creep increases as the element size decreases [31]).

### 4.7. Limitations of the Study

The authors of the presented research want to mark that the creep strain and the creep coefficient were determined with decreasing load (decrease in prestressing force). For this reason the authors estimate the error in designation strains and creep coefficient to about 10%. It is an error which allows comparison of the obtained results with others obtained in different test conditions.

### 4.8. Research Profits

The obtained results of research on rheological properties of concrete with Certyd aggregate showed low shrinkage and creep in such concrete. The research showed a better quality of the aggregate compared to other artificial aggregates analyzed in other studies. The information obtained can be a supplement to the scarce global database of results in the field of creep in lightweight concrete with artificial aggregates. They can also be an opportunity to popularize this aggregate.

Research confirmed the good quality of aggregate for concrete made with waste materials. It is of great local significance. This is important because of the dwindling supplies of natural aggregates and the large reserves of ash in Poland which remain after the production of energy from coal combustion.

The authors see the most important benefits in the possibility of using this aggregate for prestressed concrete. The lower weight of the concrete can reduce the cross-section and increase the span of the elements [4,13]. In the case of pre-tensioned prefabricates, lower weight can save transport costs.

## 5. Conclusions

The paper presents the results of tests of shrinkage, creep, and prestress losses in lightweight aggregate concrete with artificial aggregate obtained by sintering fly ash. Creep was tested under load for 539 days and shrinkage for 900 days. Based on the obtained results, it was found that:
The concrete made exhibited a lower shrinkage than that obtained from the calculations in accordance with the Eurocode 2 [36], for the assumed mixture parameters and test conditions as well as than that obtained from foreign research of concretes with other artificial lightweight aggregates; The tested concrete exhibited a very low creep coefficient in the considered period. The creep coefficient value was 0.610 and 0.537, which is 56.0 and 49.3% of the value determined from the standard [36]. It is also far less than that obtained from foreign research on concretes with other artificial lightweight aggregates. The creep rate is very fast, 95% of the creep registered after 539 days had already occurred in the first 200 days;The concrete showed clearly viscous behavior and high residual strain;The prestressing losses in the analyzed period amounted to an average of 13.0% (maximum 16.2%), which is an acceptable value and does not discriminate against concrete being used for prestressing.


To summarize the obtained test results, it was found that the tested lightweight aggregate concrete with sintered fly ash aggregate Certyd is well suited for prestressed concrete structures. Low creep guarantees low deflection increments over time. Although this concrete is characterized by a lower modulus of elasticity compared to normal weight concrete (the values obtained after 28 days were 22.1 and 22.4 GPa), several computational analyses [4,12,13] and in-situ tests [12,37] have shown that, with proper prestressing, the lowered modulus of elasticity is not problematic and does not lead to an increase in deflections.

## Figures and Tables

**Figure 1 materials-14-03895-f001:**
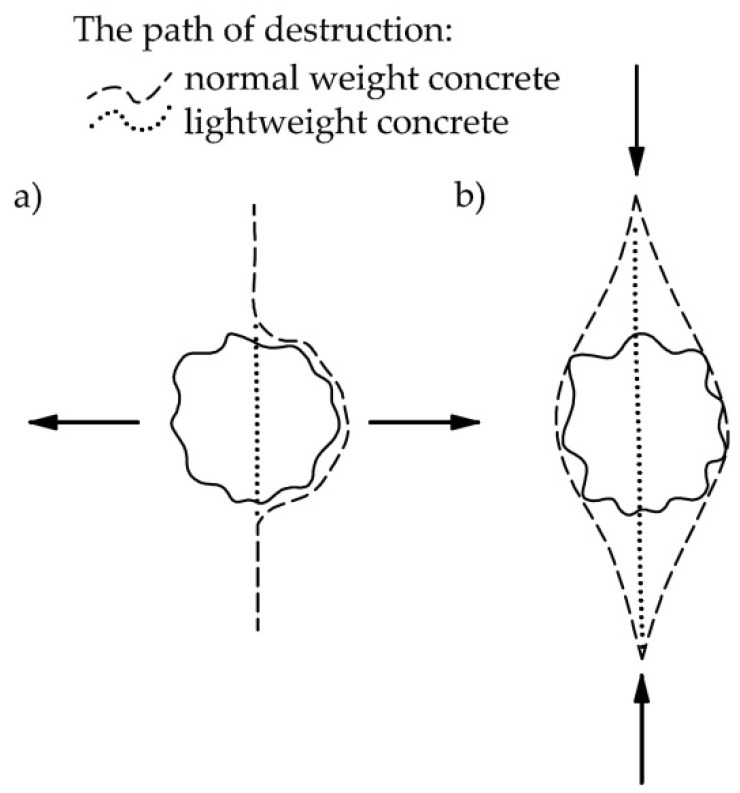
Scheme of destruction of lightweight and normal weight concrete in tension (**a**) and splitting (**b**) [4].

**Figure 2 materials-14-03895-f002:**
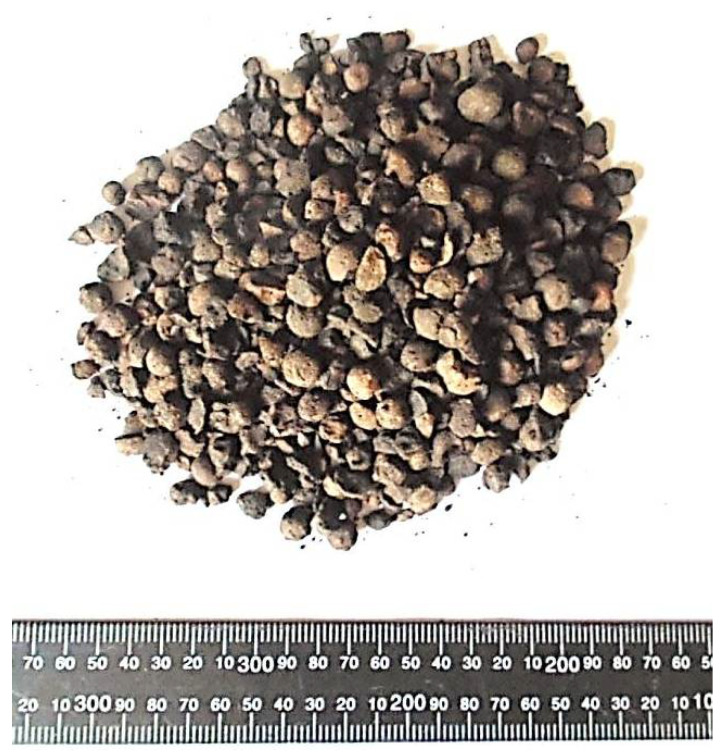
Artificial aggregate Certyd—differentiation of fraction and shape (ruler in mm).

**Figure 3 materials-14-03895-f003:**
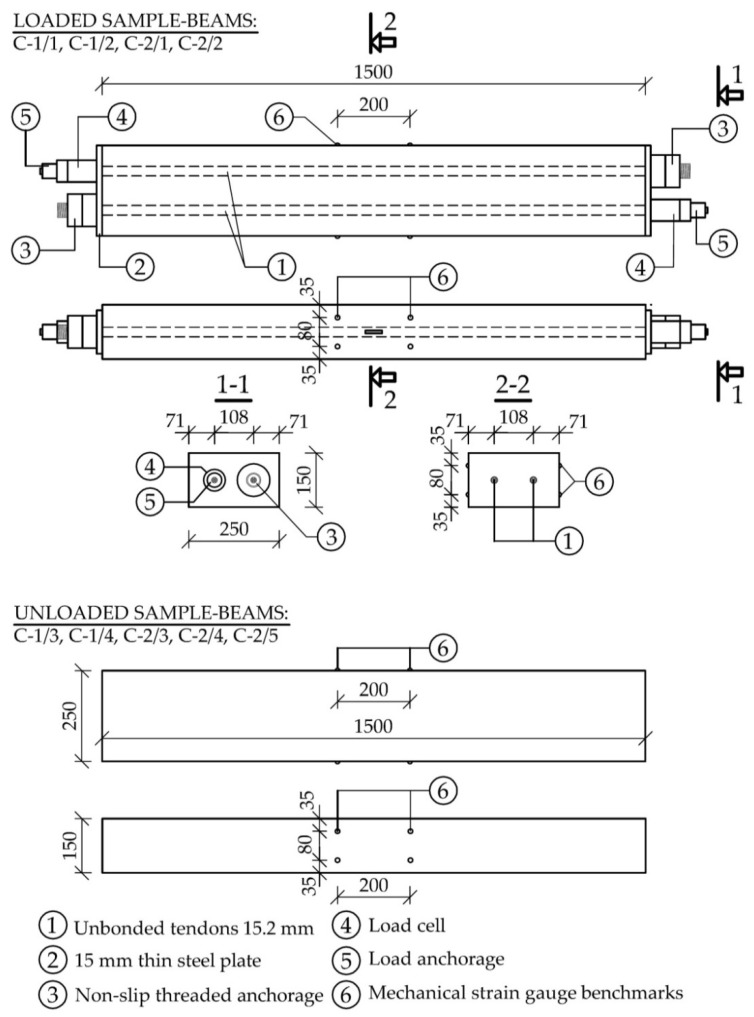
Specimens for shrinkage and creep test, scheme of prestressing and measurement bases (dimensions in mm).

**Figure 4 materials-14-03895-f004:**
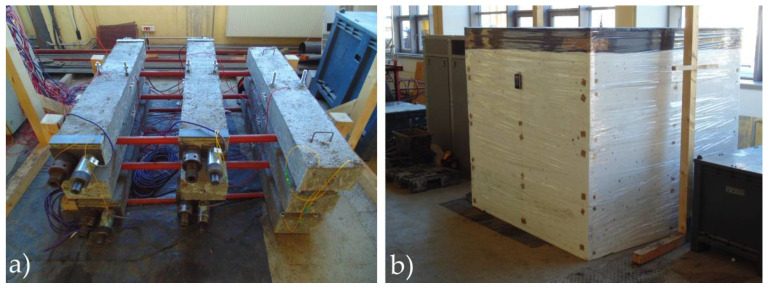
Specimens for shrinkage and creep tests on the steel frame (**a**) and air-conditioned chamber (**b**).

**Figure 5 materials-14-03895-f005:**
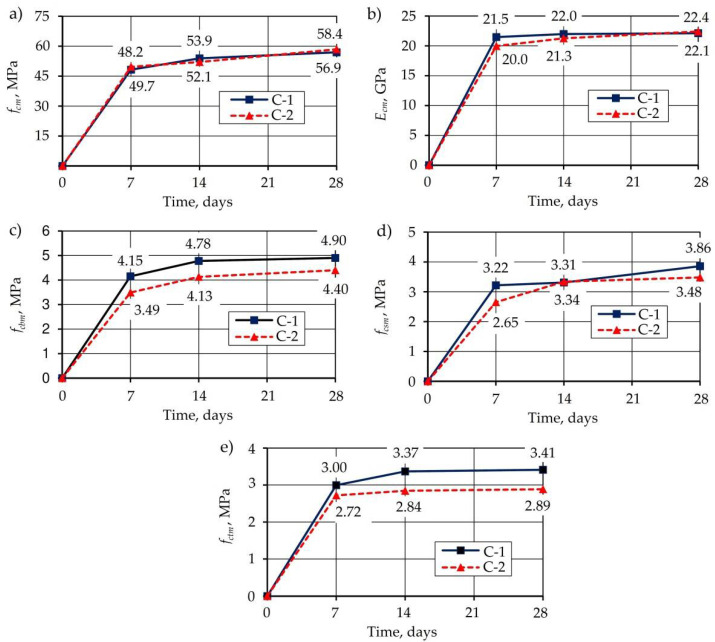
Development of mechanical properties in time: (**a**) compressive strength, (**b**) modulus of elasticity, (**c**) modulus of rapture, (**d**) axial tensile strength, and (**e**) splitting strength.

**Figure 6 materials-14-03895-f006:**
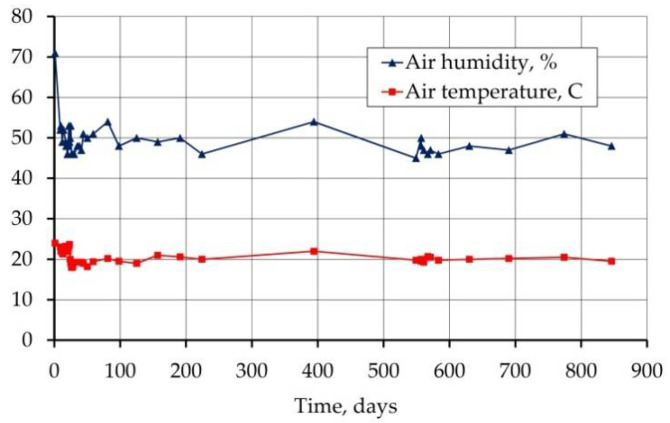
Air temperature and humidity in the air-conditioned chamber.

**Figure 7 materials-14-03895-f007:**
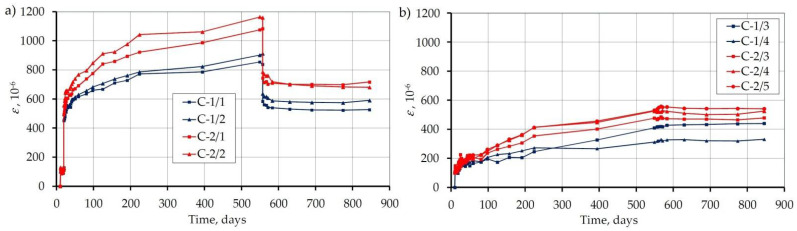
Strain of loaded (**a**) and unloaded (**b**) specimens.

**Figure 8 materials-14-03895-f008:**
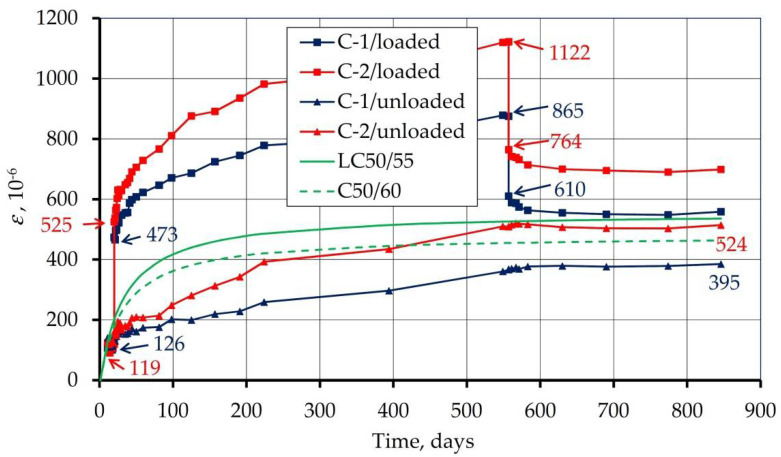
Average strains of loaded and unloaded specimens, as well as shrinkage according to the standard [36] for LC50/55 and C50/60 class concrete.

**Figure 9 materials-14-03895-f009:**
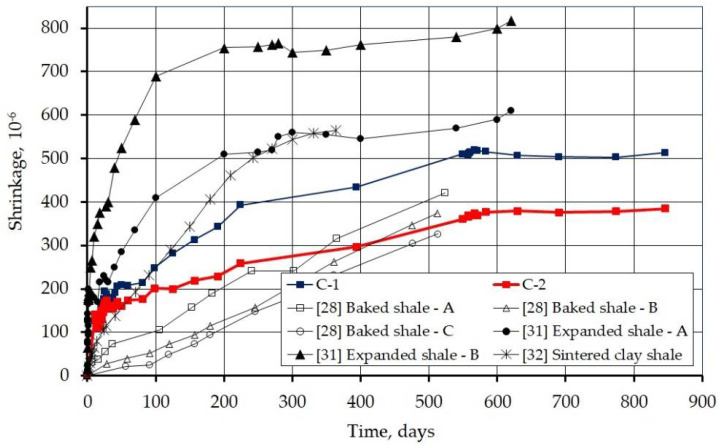
Measured shrinkage strains compared with the results of other studies.

**Figure 10 materials-14-03895-f010:**
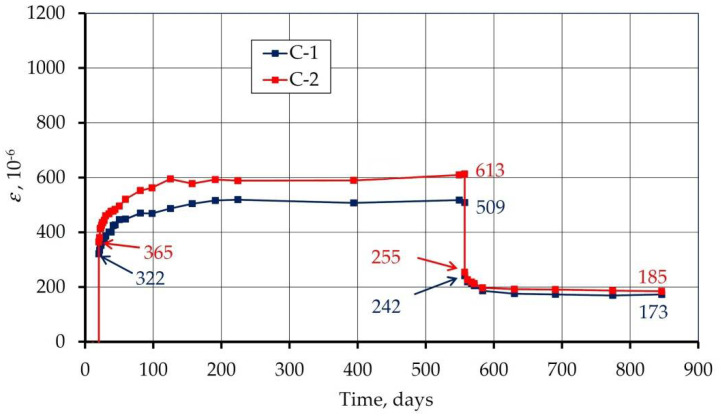
Creep strain (the difference between strains of loaded and unloaded specimens).

**Figure 11 materials-14-03895-f011:**
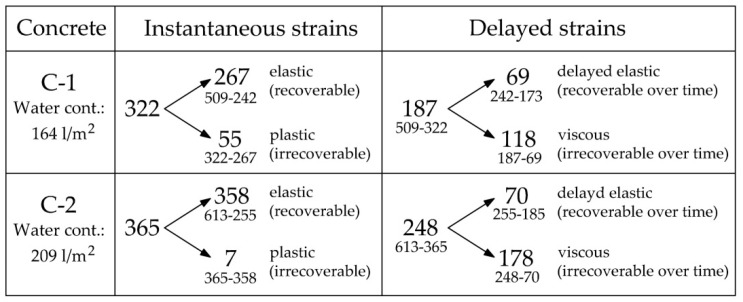
Concrete strains under the load in 10^−6^ (four components were separated: elastic, plastic, delayed elastic, and viscous strain).

**Figure 12 materials-14-03895-f012:**
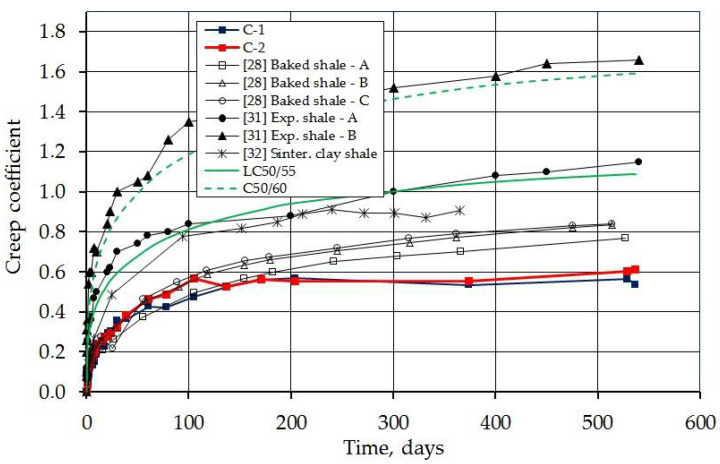
Development of creep coefficient over time (measured, calculated from the Eurocode 2, and taken from foreign studies).

**Figure 13 materials-14-03895-f013:**
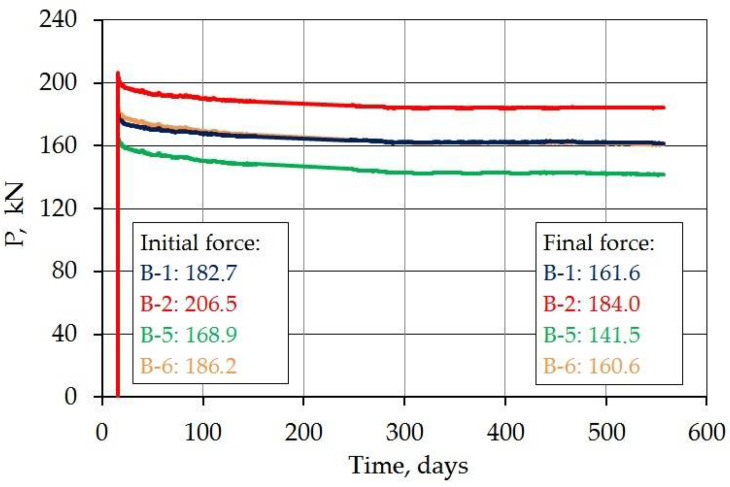
Change of force in prestressing tendons over time.

**Table 1 materials-14-03895-t001:** List of published light concrete creep tests with the most important parameters.

Research	Lightweight Aggregate	Concrete Strength MPa	Loading AgeDay	Samples Number	Load Time Days
Best and Polivka 1959 [28]	Baked shale	20.7	-	4	520
		34.5		3	
Pfeifer 1968 [29]	Expanded blast furnace slag	20.7 34.5	7	-	730
	Expanded shale produced in a rotary kiln				
	Expanded shale produced on a sintering grate				
	Expanded clay produced on a sintering grate				
Lopez, Kahn and Kurtis 2004 [30]	Expanded shale	55.0 69.0	16	8	620
Lopez 2005 [31]	Pre-soaked extended shale	Shrinkage and creep were not separated, the creep is provided based on the DIC technique			
Wendling, Sadhasivam and Floyd 2018 [32]	Expanded shale	28.0	1 28	4	365
Wang, Li, Jiang, Wang, Xu and Harries 2020 [33]	Expanded clay	25.1 ÷ 24.8	28	16 large scale RC and PT beams	20–30 years
	Sintered pulverized fuel ash				
Lukin, Popov and Lisyatnikov 2020 [34]	Expanded clay	10÷60	28	-	700
	Expanded perlite				
	Agloporite				

**Table 2 materials-14-03895-t002:** Values of basic parameters of Certyd aggregate, based on [35].

Feature	Code	Fraction			
		0/2	2/4	4/8	8/16
Bulk density, kg/m^3^	PN-EN 1097-3	930–990	600–630	650–750	740–750
Grain density, kg/m^3^	PN-EN 1097-6	-	-	1350–1430	1350–1430
Water absorption after 12 h, %	PN-EN 1097-6	-	-	17	16
Crush resistance, MPa	PN-EN 13055-1	-	-	6–10	6–8
Frost resistance, %	NP-EN 13055-1	-	-	≤1	≤1

**Table 3 materials-14-03895-t003:** Composition of concrete mixtures made.

Component		C-1	C-2
		kg/m^3^	kg/m^3^
Cement CEM I 42.5 N		409	419
Aggregate Certyd (4–12 mm)		775	802
Sand		682	703
Water		164	209
Additions	BV 18	3,7	3,8
	SKY 686	3,7	3,8
Weight of wet mixture		2039	2142
Concrete density		1810	1820
W/C		0.41	0.51

**Table 4 materials-14-03895-t004:** Values of initial compressive stresses in specimens.

Concrete	Specimen	Initial Stress, MPa
C-1	C-1/1	9.7
	C-1/2	11.0
	C-1/3	-
	C-1/4	-
C-2	C-2/1	9.0
	C-2/2	9.9
	C-2/3	-
	C-2/4	-
	C-2/5	-

**Table 5 materials-14-03895-t005:** Values of concrete mechanical properties for class LC50/55 according to [36] and test results.

Feature	Unit	Required	Test Results	
			C-1	C-2
*f_lcm_*	MPa	58	56.9	58.4
*E_lcm_*	GPa	25.3	22.1	22.4
*f_lctm_*	MPa	3.67	3.86	3.48

## Data Availability

The presented research was conducted in Building Materials and Structures Research Laboratory of Cracow University of Technology, Warszawska 24 Street, Cracow 31-409, L-12@pk.edu.pl.

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
