# Peer review of "Experimental Evaluation of Shrinkage, Creep and Prestress Losses in Lightweight Aggregate Concrete with Sintered Fly Ash"

_materials, 2021, doi:10.3390/ma14143895_

Round 1
Reviewer 1 Report
After going through the manuscript " Experimental Evaluation of Shrinkage, Creep and Prestress Losses in Lightweight Aggregate Concrete with Sintered Fly Ash", I would give my comments below.
This manuscript does not correspond to the high-quality criteria of the journal of Materials. Therefore, my decision is rejection this manuscript.
- Introduction part needs to a schematic to describe the procedure of your research.
- The methodology section is not well organized for the readers to understand the concept.
- The language used in the manuscript can be more specific to the scope and aim of the study.
- Materials section needs characterization tests such SEM, XRD, and ….
- the manuscript needs more tests and analyses.
-The authors should review the other new investigation on their study way in the introduction part and finally note the novelty of the article. The introduction part needs to develop.
- Section “Prestress force values in time” is poorly written. More details on quantities should be provided.
- Explain the results obtained from the SEM analyses more thoroughly.
- Illustrate better the beam strains technique and the results obtained.
- Conclusion is very long. should be revised.
Reviewer 2 Report
This manuscript is about experimental evaluation of shrinkage, creep and prestress losses in lightweight aggregate concrete with sintered fly ash.
Some suggestions, the authors should consider improving the quality of manuscript:
Abstract: Add something about the benefits results of the research.
-Please add a list of those materials which have already been used in previous studies for same purpose and now this type of material will replace. [Please add in Section 4]
-Please elaborate a clear research gap in some lines may be pattern can be
- Introduction
- Background
2.1 Properties of Lightweight Concrete
2.2 Creep and Shrinkage Research
2.3 Research Gap
- Aggregate, Concrete Mixture, Research Program…It can be ‘Materials and Methods’
- Results and Discussion…. also correct…..’and’ missing
- Conclusions
Aggregate, Concrete Mixture, Research Program
Change heading wth standard formate- ‘Materials and Methods’
A comprehensive research framework missing- to follow the research is steps are missing. Add framework-flowchart and write this section in stepwise pattern.
Step.1, Step.2 …..[Please add]
Please add details of atleast 5-6 materials which were used in previous studies and how authors got an idea to use this specific material. Further also explain how it was proved that this material has certain properties to replace existing materials. [Please add]
Results
Please compare the results of produced concrete within existing published research. Line or bar charts can be added. [Please add]
Discussion:
This section must contain implications for research, practice and/or Field: Does the paper identify clearly any implications for research, practice and/or society? Does the paper bridge the gap between theory and practice? How can the research be used in practice (economic and commercial impact), to influence technical policy, in research (contributing to the body of knowledge)? Add something for field professionals. [Please add]
Limitations of the study:
Please add as heading 6 about the limitations of the study.
Reviewer 3 Report
The manuscript “Experimental Evaluation of Shrinkage, Creep and Prestress Losses in Lightweight Aggregate Concrete with Sintered Fly Ash”is well organized and do a fruitful work. As the creep testing is long circle testing, authors carrying out the creep test up to 539 days and concrete shrinkage for 900 days
The caption of figure 2 is same with figure 1, and figure 2 and figure 1 can be merged.
The ruler should be given in Fig.3
The caption for figures is too simple especially for Fig.11 and Fig.12
Figs 10-12 can be merged
The mechanism about the shrinkage and creep after long time treatment should be discussed. If it is possible, it is better to give same microstructure or phase changing during long time of under prestressing load.
The creep study is few for lightweight aggregate concrete, authors can add some references about creep and shrinkage for comparison, a table for comparing is advised.
Any way, this paper can be accepted after minor revision.
Round 2
Reviewer 1 Report
The authors considered the comments of the reviewer. The revised manuscript is significantly improved. However, because some parts are still obscure, the explanations in Figure 2 are not sufficient and need to be analyzed and referenced. The novelty of the work remains unclear; authors should describe the novelty of the work and the difference between this work with other similar research.
Reviewer 2 Report
This manuscript can be accepted in current form.
Author Response
Dear Reviewer
Thank you for your valuable comments.
Some wording has been further improved.
Kind regards
Rafał Szydłowski